# Coating of Layer-by-Layer Assembly Based on Chitosan and CMC: Emerging Alternative for Quality Maintenance of Citrus Fruit

**Chenyu Niu [1], Lingling Liu [1], Amr Farouk [2], Cunkun Chen [3] and Zhaojun Ban [1,*]**

[1] Zhejiang Provincial Key Laboratory of Chemical and Biological Processing Technology of Farm Products, School of Biological and Chemical Engineering, Zhejiang University of Science and Technology, Hangzhou 310023, China; 212203817034@zust.edu.cn (C.N.); llliu@zust.edu.cn (L.L.)

[2] Flavor and Aroma Chemistry Department, National Research Centre, Cairo 12622, Egypt; af.mansour@nrc.sci.eg

[3] National Engineering Technology Research Center for Preservation of Agriculture Product, Institute of Agricultural Products Preservation and Processing Technology, Tianjin Academy of Agricultural Sciences, Tianjin 300384, China; cck0318@126.com

[*] Correspondence: banzhaojun@zust.edu.cn; Tel.: +86-571-8759-0393

**Abstract:** Citrus fruits are susceptible to microbial infection during storage and transportation, leading to weight loss, quality deterioration and even physiological decay. In this study, a layer-by-layer (LbL) self-assembly edible coating based on chitosan and Sodium carboxymethyl cellulose (CMC) was prepared using electrostatic deposition. Postharvest citrus fruits (*Citrus reticulate* cv. 'Chunxiang') were coated either with the LbL coating, which exhibited an increased transmission rate of water vapor, or with single chitosan coating. The data of some physiological indexes of citrus coated with LbL coating and single coating were compared after being stored at 13 ± 2 °C and at relative humidity (RH) at 80–85% (shelf-life condition) for 10 days. Results demonstrated that the LbL deposited coating was effective at maintaining morphological and microstructural attributes, increasing the surface brightness, firmness and the content of titratable acid and ascorbic acid, as well as reducing the weight loss after storage by about 0.8%. Results in the present study indicated that the LbL edible coating could have the potential to maintain postharvest citrus quality during storage.

**Keywords:** chitosan; CMC; FTIR; SEM; postharvest citrus; quality





## 1. Introduction

Compared with traditional plastic film, bio-based film has good biodegradability, so in recent years bio-based film preservation packaging has become a research hotspot [1]. Polysaccharide-based edible film is usually made of starch, chitosan, cellulose and their denaturation products. The edible layer of polysaccharide-based film can not only delay the aging of fruits and vegetables, maintain the overall quality of fruits and vegetables and extend the shelf life, but also provide a certain degree of antibacterial activity and degradability [2].

Chitosan, a biodegradable polymer produced by the deacetylation of the natural polysaccharide chitin [3], possesses intrinsic antimicrobial activity towards bacteria, yeast, and molds [4]. Therefore, chitosan can be processed into edible composite membranes based on these properties. For example, Patricia Zimet et al. discovered that a composite membrane made of chitosan and carboxymethyl chitosan had good bacteriostatic performance [5]. However, the physical properties of chitosan, such as solubility, gelation, and affinity, limited its application when researchers sought to form an optimized edible coating, resulting in an uneven distribution of the coating, hindering its performance [6].

CMC is polyanionic when dissolved in an aqueous solution, due to its carboxylic substituent, providing a uniform and stable matrix. In addition, CMC can form hydrogen bonds with the cuticle of orange peel to maintain a high degree of structural integrity.

Due to the limitations of their functions, single-coating materials often cannot meet the practical needs of diversification. In this case, greater interest in multicomponent coatings was required. At present, rationally designed multi-component edible coatings are being sought to provide new materials with well-controlled, beneficial performance [7]. The LbL electrostatic deposition technique originated in materials science and has extensive applications [8]. This approach is based on the alternate deposition of oppositely charged polyelectrolytes and is designed to effectively control material performance and function. The application of LbL edible coatings, which is based on alginate and chitosan, was found to significantly improve the physiological and microbial quality of fresh-cut melons with no addition of any active agents [9]. It has been reported that multilayer coating can form a barrier on the fruit surface. The presence of this barrier can reduce water loss in treated fruits and maintain the total phenolic content and pH during storage. The coating reduces the fruit's respiration rate and affects its ripening process. Therefore, multilayer coating can extend the shelf life of fresh annona [10]. Chitosan was found to provide mandarins with improved gloss and good firmness [11,12]. However, single chitosan coating will still reduce the performance of fruit due to the influence of water vapor transmittance on the surface [13]. In addition, chitosan dissolved in a slightly acidic water solution is generally in a polycationic state. Therefore, we can use the method of layer-by-layer deposition to form density-induced double-layer usable films through electrostatic interaction between CMC and chitosan, which can make up for the defects of single-layer films [7]. *Citrus reticulate* cv. 'Chunxiang' is a new hybrid citrus variety that has been introduced in recent years, with good characteristics of large fruit, a sweet and clear taste, few seeds, high sensory quality and high and stable yield, as well as good economic benefit (in China). After harvest, citrus fruits are prone to physiological disorders and microbiological decay. Normally, citrus fruits are coated with commercial waxes on the peel to decrease water loss and shrinkage and enhance the gloss of the fruit. However, the wax coatings may also debase fruit quality, as they may restrict gas exchange through the peel, thereby resulting in the development of anaerobic conditions in the internal atmosphere of the fruit, the accumulation of ethanol, and the presence of off-flavors [14]. In the experiments of Saberi, B., they compared the effects of edible composite coatings with commercial waxes on oranges and came to the same conclusion [15].

The study aimed to study the protective effect of chitosan/CMC double-layer edible coating on the postharvest quality of citrus fruits. The representative quality indexes of citrus fruits, including weight loss, total soluble solids (TSS) and titratable acidity (TA), were determined. This study provides a theoretical basis for the application of this edible double-layer coating in fruit preservation.

## 2. Materials and Methods

### 2.1. Preparation of Edible Coating Formulations

#### 2.1.1. CMC Coating

CMC powder (Aladdin Biochemical Technology Co., Ltd., Shanghai, China) was dissolved in sterilized water upon stirring at 50 °C for 20 min to obtain a 1.5% (*w/v*) solution.

#### 2.1.2. Chitosan (CH) Coating

Chitosan powder (Aladdin Biochemical Technology Co., Ltd., Shanghai, China) was dissolved in sterilized water that included 1% (*v/v*) of acetic acid upon stirring at 50 °C to obtain a 1.5% (*w/v*) solution.

### 2.2. Preparation and Characterization of LbL Film

Chitosan (1.5%, *w/v*) was poured into a glass Petri dish measuring 6 cm in diameter and dried overnight in a constant temperature and humidity chamber (23 °C, RH 65%).

For the chitosan/CMC composite LbL film, first 5 mL of 1.5% CMC solution was poured into a 6 cm diameter Petri dish, and then it was dried overnight in a constant temperature and humidity chamber (23 °C, RH 65%). The second layer was then cast with 5 mL of 1.5% chitosan solution [16].

The properties of LbL film were determined using the method of Poverenov et al. [9]: Fourier transform infrared (FTIR) spectroscopy was used to determine the infrared absorption spectrum of the film at 4000–400 cm$^{-1}$. The microstructure and thickness of the film was determined with scanning electron microscopy (SEM). The mechanical properties (breakdown strength and elasticity) of the film were measured using a texture analyzer. The water vapor permeability (WVP) measurement was carried out using the method of FTIR spectra of the prepared films, recorded between 400 and 4000 cm$^{-1}$ with one hundred scans averaged with a resolution of 4 cm$^{-1}$ (Bruker Tensor 27 FTIR Spectrometer). The characterizations of LBL membranes had been evaluated by our group in previous experiments and publication [16].

### 2.3. Coating on Citrus

#### 2.3.1. Plant Materials

The *Citrus reticulate* cv. 'Chunxiang' used in the experiment was provided by Guotiantian Citrus Cooperative in Kecheng District, Quzhou City, Zhejiang Province, China. The citrus fruit was collected in a well-managed orchard and was immediately sent to the experimental site within 2 h without any treatment. Fruits with uniform size, no pests or diseases, and no mechanical damage were selected by manual sorting for further analysis.

#### 2.3.2. Application of Edible Coatings

Citrus fruits were divided into a control group (CT), a single chitosan coating group (CH) and a chitosan/CMC self-assembled LbL coating group (CC), among which the CT group was treated with distilled water. The CMC coating was evenly applied to the citrus surface by hand with a brush, dried at room temperature for 0.5 h, then the chitosan coating was applied, and dried at room temperature until the fruit surface was completely dry [12]. Each treatment group included 25–30 mandarins kept in cardboard boxes. The mandarins were stored for 20 days at 13 ± 2 °C and a relative humidity (RH) of 80–85% (shelf-life conditions).

### 2.4. Fruit Quality Attributes

#### 2.4.1. Morphological and Microstructure Characteristics

Five fruits were randomly selected from each group, and photos were taken of the whole and crosscut fruits. Epidermis sections of approximately 1 cm by 1 cm in size were cut from the surface of the citrus and divided into four pieces with a thickness of 1 mm. The sample was placed in 2.5% glutaraldehyde solution and allowed to stand for 12 h and was then removed with 0.2 M phosphoric acid buffer (PBS) and rinsed three times, then 2% argon acid was added and fixed for 1 h before being washed three more times with PBS. Subsequently, the samples were successively placed in 30%, 50%, 70%, 80%, 90% and 95% ethanol for dehydration with immersion for 15 min. Then, anhydrous ethanol was added twice for dehydration and immersion for 20 min each time. The samples processed by freeze-drying were mounted on metal stubs and coated with gold using a Polaron sputter coater. The scanning electron micrograph (SEM) was recorded using the scanning electron microscope SU8010 (Hitachi, Tokyo, Japan), [17].

#### 2.4.2. Fruit Color, Firmness and Weight Loss

The color of the fruit peel was determined in five randomly removed fruits for each group, and was measured on opposite sides of the peel of the fruit, using a colorimeter to measure changes in citrus color (L*) before and after storage [18].

The firmness of both symmetrical sides of the fruit was measured using a TA-XT2i firmness analyzer with at least 5 fruits in each group. The measurement used the P5 probe with a diameter of 5 mm, a depth of 10 mm and a speed of 0.5 mms$^{-1}$ [19].

Fruit weight loss was evaluated by weighing the same fruit before and after storage, and the data were the means from 15 fruits $\pm$ standard deviations (SD). The results were presented as % [7].

### 2.4.3. Fruit Total Soluble Solids (TSS), Titratable Acid (TA) and Ascorbic Acid (AA)

The fruit juice was squeezed through four layers of gauze and mixed for use to determine the TSS and TA content of the citrus [20]. TSS and TA contents in the juice were determined with a PAL-BX/ACID F5 sugar-acid meter. The AA content was measured using a kit (Ascorbic Acid TEST Kit- lot- HI3850, Hanna Instruments, Woonsocket, RI, USA) based on the 2,6-dichloroindophenol method [21]. Each determination was carried out on five citrus fruits and in three technical replications.

### 2.5. Statistical Analysis

The means and SD of each determination were calculated using Microsoft Excel software, and Duncan's multiple difference significant analysis was performed using the SPSS (R26.0.0.0) statistical software processing system. Using a one-way analysis of variance, $p < 0.05$ indicated that the difference was significant at the 0.05 level. The vertical line in the figure represented the standard error, and the same letter indicated that the difference was not significant. Three biological replications and three technical replications were carried out in the whole experiment.

## 3. Results and Discussion

### 3.1. Properties Characterization of the Films

The FTIR spectra of chitosan powder, chitosan film, CMC powder and chitosan/CMC LbL films were measured (Figure 1). In accordance with previous authors [9], chitosan films demonstrated characteristic bands at 1637 and 1570 cm$^{-1}$ (assigned to an amide bond); 3400–3500 cm$^{-1}$ (assigned to O-H and N-H stretching); and 900 and 1150 cm$^{-1}$ (assigned to pyranose rings and amino groups).

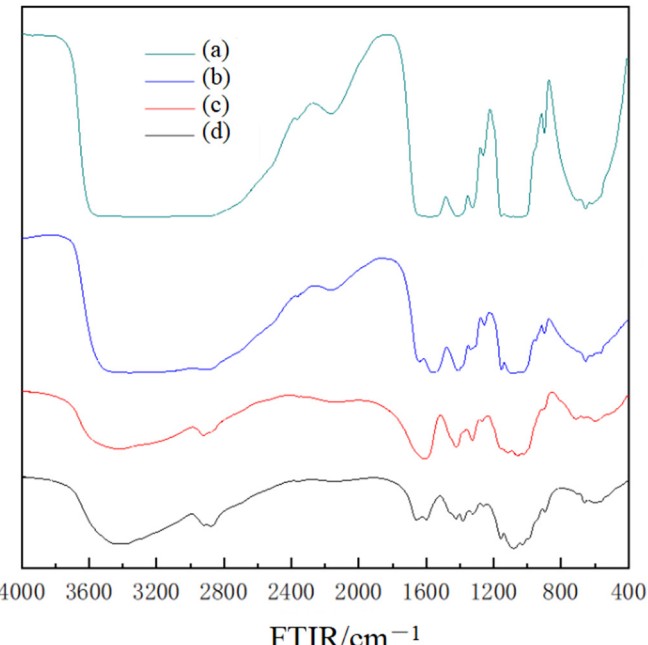

**Figure 1.** This FTIR spectra of the prepared films. (**a**) Chitosan/CMC LbL film; (**b**) chitosan film; (**c**) CMC powder; and (**d**) chitosan powder.

The absorption intensity of the prepared film at 3300 cm$^{-1}$ was stronger than the absorption intensity of the responding powder, and the absorption intensity of the LbL film was stronger than the single-layer film, which indicates that the hydrogen bonds were formed between the chitosan/CMC and water during the preparation process of the LbL film. The peak structure of the chitosan film was largely similar to that of the chitosan powder, and the characteristic peaks were at around 3300, 1637 and 1570 cm$^{-1}$, indicating that the ethylated chitosan only dissolved without other reactions, and the basic skeleton of the molecule was unchanged during the dissolution process. Approximately 1608 and 1080 cm$^{-1}$ indicated, respectively, the CMC C=O and C-O stretching vibration characteristic peaks, and the composite film measurements were approximately 3366, 1600 and 1063 cm$^{-1}$. Results demonstrated that the absorption peaks appeared, indicating that the LbL composite film had the characteristic spectra of both of the two materials. The solution base materials could interact with each other. In the experiment of Poverenov et al. (2014), gelatin and chitosan could interact with each other through LbL technology and showed compatibility between the two substances [12].

In the previous experiment [16], the SEM results indicated that the surface of the prepared chitosan film was highly smooth and compact. The LbL composite film had a regular cross-sectional structure, and the two membranes were closely connected. It was demonstrated that the chitosan cross-section structure was closely arranged, and the structure was regular, while the CMC had a network structure with a relatively poor arrangement. The chitosan/CMC LbL film showed a rough surface with small wrinkles. After the addition of CMC nanoparticles, the surface of the chitosan/CMC film became rougher because of the embedded and well-dispersed CMC in the chitosan matrix [22].

Based on the previous results reported in our group [16], the thickness of the composite film was less than two times the thickness of the single-layer film. Additionally, the disruptive strength of the LBL membrane was about 1.25 times that of chitosan membrane. The elasticity of LBL films measured by us was almost the same as that of chitosan films. The WVP of the chitosan membrane was 47.36 ± 0.00, and that of the LBL membrane was 82.55 ± 4.64. It was obvious that the composite coating (LBL film) had higher water vapor permeability. We speculated that the swelling of the chitosan membrane or the cross-linking between chitosan and CMC in the LBL membrane promoted the transport of water molecules into the membrane.

### 3.2. Citrus Quality Attributes

Color is considered one of the most important aspects of fruits and vegetables when it comes to visual appeal to consumers [23]. It can be seen from Figure 2 that the citrus in the CT group had darker colors, reduced brightness, fibrosis, and partial water loss, indicating that the coating film could reduce the loss of spring fragrance and lemon moisture and extend the shelf life.

Figure 2 shows that the coated citrus had a brighter color and better appearance quality. In addition, the size and number of stomata in the cuticle of the citrus were also affected by coating. SEM showed that the orange peel was smoother after coating. As Figure 3 shows, both coatings could fill the stomata of the citrus, reducing the amounts of stomata and porosity and inhibiting the respiration and transpiration rates. At the later stages of storage, the surface of the fruits in the CH group was rough, while in the CC group there was less degradation, indicating that the CC composite film increased the barrier performance and protection ability against pathogens. Both Figures 2 and 3 show that the closed stomata had a positive effect on prolonging the shelf life of citrus fruit.

The color of the fruit peel can be used as a criterion for evaluating fruit ripeness and is an important indicator for determining fruits' commerciality. The color difference L* indicates brightness, and the larger the L value is, the higher the brightness of the fruit is. After 20 days of storage under the same conditions, the L* values of the three groups of citrus measured from high to low were 74.684 (CH), 74.343 (CC) and 73.97 (CK). Figure 4a shows that the L* values of citrus in the CC group and CT group were higher than the L*

value of citrus in the CT group, indicating that the LbL composite coating improved fruit morphological characteristics to a certain extent, but had no significant effect on the color of citrus. Armon H et al. found that the chitosan and CMC composite coating improved the color and appearance of grapefruit, and the shell polysaccharide coating had a good effect on peaches, papaya and strawberries [12].

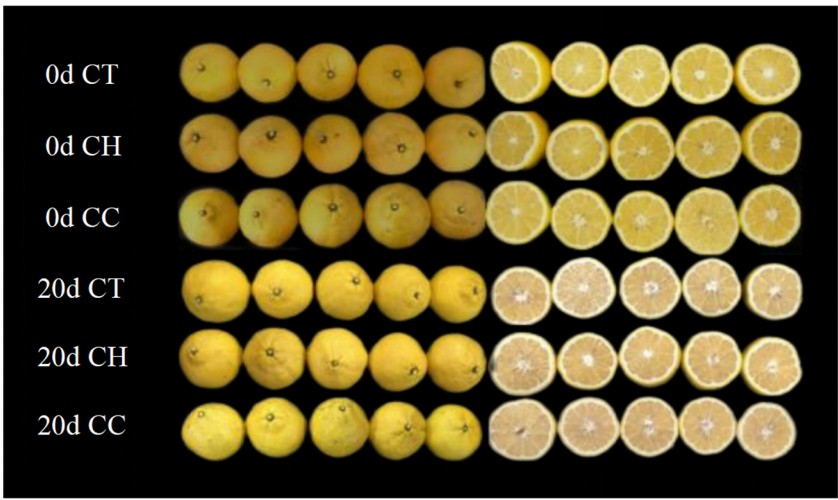

**Figure 2.** Morphology of citrus coated with CT, CH and CC before and after storage (CT, control experiment; CH, single chitosan coating group; CC, chitosan/CMC self-assembled LbL coating group.).

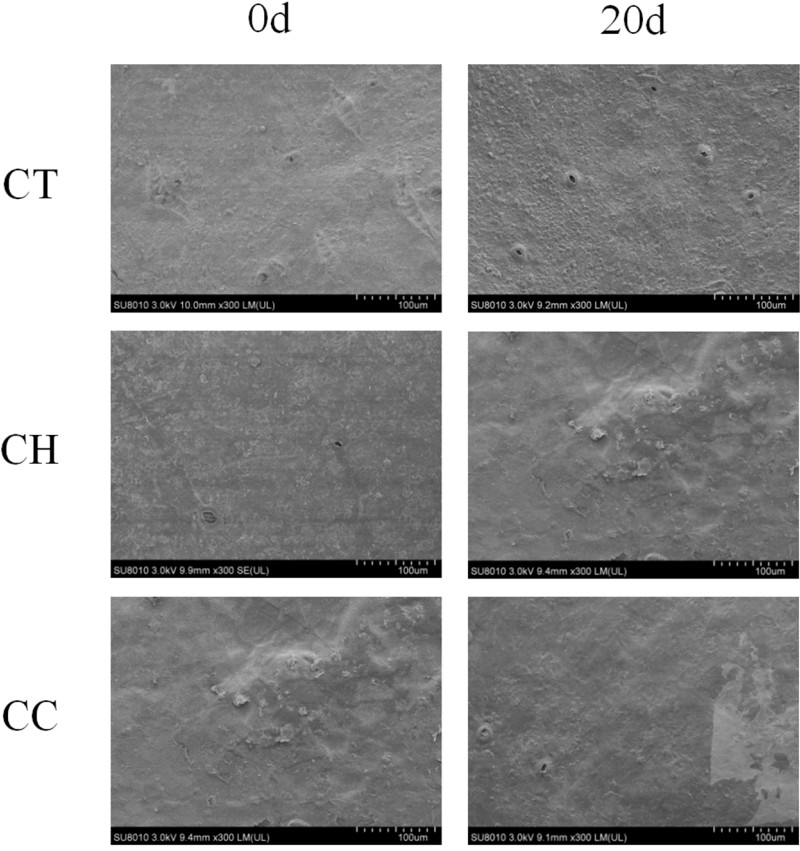

**Figure 3.** SEM graphs (×300) of citrus coated with CT, CH and CC before and after storage.

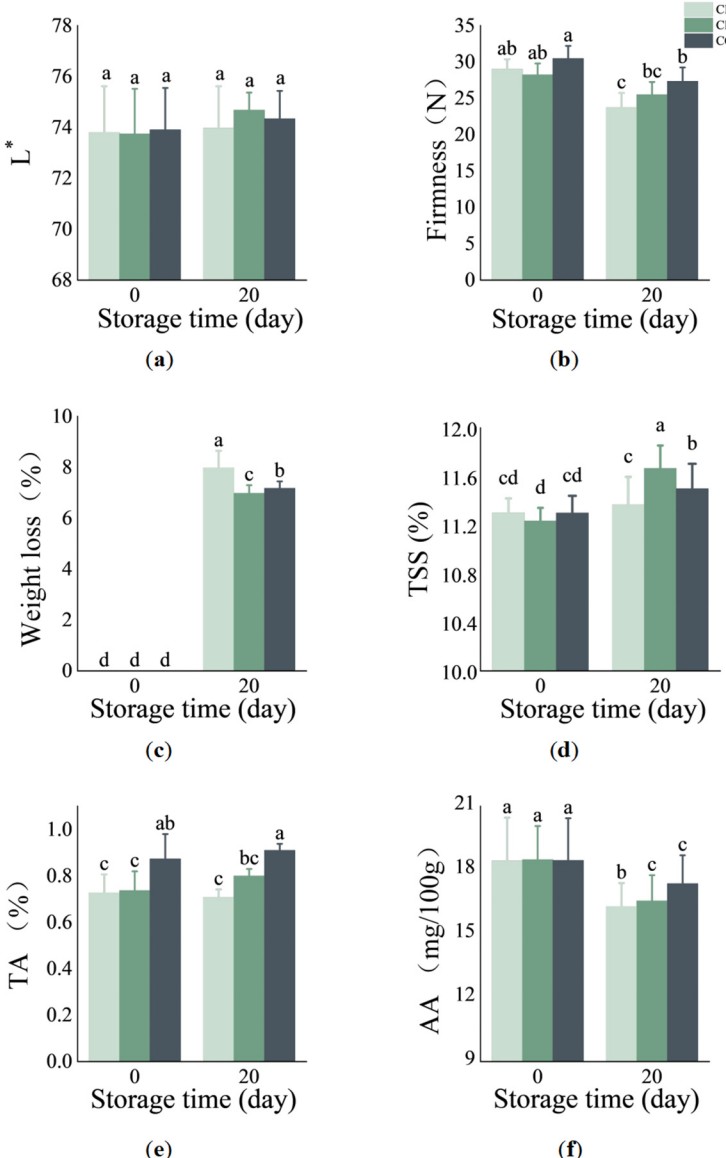

**Figure 4.** (**a**) L* value; (**b**) firmness; (**c**) weight loss rate; (**d**) TSS; (**e**) TA; and (**f**) AA content of citrus before and after storage. Different letters indicate significant difference in all treatments during storage period ($p < 0.05$).

Fruit firmness is an important basic index to evaluate fruit quality and an important parameter for measuring the characteristics and commodity of citrus fruit storage and transportation. As Figure 4b shows, the firmness of the citrus fruits stored for 20 d with three treatments, ranked from high to low, was in the following order: CC, CH and CT. The results show that both coatings could improve the firmness of citrus fruit. The LbL composite coating could prolong the storage period of citrus and provide a new approach for long-distance transportation and the long-term storage of citrus fruit. This result is consistent with the results of PEM coating (synthesized with CMC and chitosan) performed well on preserving apple quality [24], and the fresh-cut melon model [25].

Studies have shown that the chitosan coating can significantly inhibit water loss in fruits and vegetables such as longan [26] and okra [27]. In Yan, J.'s experiment, the coating could also improve fruit quality and maintain the fresh fruit weight during storage [28]. It was consistently confirmed that in Figure 4c the coating had a significant effect on the weight loss rate of citrus after storage for 20 days ($p < 0.05$). Both the CH group and the CC group decreased the weight loss after citrus storage significantly ($p < 0.05$), showing that

the coating treatment can reduce the weight loss and improve the value of the commodities of citrus fruit. Nevertheless, in the present study, the weight loss when coated with CH was the lowest among all groups, which may be due to the high WVP of the prepared LbL composite film, which increased the weight loss. Overall, the chitosan/CMC LbL composite coating delayed the decline in weight loss during storage, and extended the shelf life of citrus fruit.

The change in TSS and TA content is the key factor that affects the taste of fruit and is the main index from which to judge the fruit ripeness and sensory quality. As Figure 4d shows, both the CH and CC groups significantly increased the TSS content in comparison to the control, with each increasing by 0.30% and 0.13%, respectively. TA refers mainly to organic acid (malic acid, citric acid and tartaric acid) in the fruit. The change in the content of the organic acid could enormously affect the acidity and taste of the fruit. Generally, fruit after the development of organic acid content had the highest content, and its content declined with the ripening and senescence process. The main reason was that the decomposition of organic acids was greater than the synthesis from the matrix involved in such processes as respiration and gluconeogenesis in postharvest [29]. As Figure 4e shows, the TA content of both the CH and CC fruits was higher than the TA content of the control, indicating that the coating film could effectively inhibit the decomposition of TA in the citrus fruit and maintain a good taste. Previous studies have shown that coating with the chitosan coating could delay the rate of TA decline and have a positive effect on prolonging the shelf life of pomegranate [30]. It has also been previously reported that the LbL coating had a significant effect on the TA content of pomegranate arils [31] and strawberries [32]. LbL coating can reduce the respiration rate of fruits, thus delaying the utilization of organic acids to prolong the shelf life of fruits [15]. AA is strongly reductive, belonging to the important non-enzyme-induced reactive oxygen removal system, with a free radicals and antioxidant effect in fresh fruits and vegetables. Nevertheless, it is sensitive to environmental stress stimuli. During the storage period, it was noted that the AA content declined in all samples (Figure 4f). The CC groups maintained a higher content of AA after storage. The content of citrus VC in the CC group was 1.071 mg/100 g higher than that in the CT group, indicating that the composite film formed by chitosan and CMC had a positive effect on the preserving effect of citrus, which could delay the oxidation process of AA in citrus fruits and maintain the good edible quality and storage properties of citrus fruits. Results from the present study were consistent with the findings of Shi S [26] regarding the maintenance of AA in longan by a chitosan nanoscale silicon coating. Previous studies have reported a retention of AA in strawberries coated with chitosan enriched with peony extract [33], and in pepper coated with chitosan enriched with cinnamon oil [34]. The AA values observed in this study agree with those reported by Chen [35] for mandarin.

**4. Conclusions**

This study examined the effect of an LBL membrane applied to citrus on the basis of previous experiments. The results demonstrated that the chitosan/CMC LbL coating helped to maintain the quality of postharvest fruit of *Citrus reticulate* cv. 'Chunxiang', by improving the morphological and microstructural attributes, increasing the surface brightness, firmness and the contents of titratable acid and ascorbic acid and by reducing the weight loss after storage. In summary, an LbL coating based on chitosan/CMC can potentially be used to extend the shelf-life and maintain the quality of postharvest citrus.

**Author Contributions:** Conceptualization, L.L.; methodology, A.F.; data curation, C.C.; writing—original draft preparation, C.N.; writing—review and editing, C.N. and Z.B. All authors have read and agreed to the published version of the manuscript.

**Funding:** This work was supported by the National Natural Science Foundation of China [grant number 32172268]; the Key Research and Development Program of Zhejiang Province [grant number 2022C04039, 2021C02015]; the Research Project on Postgraduate Teaching Reform of Zhejiang Uni-

versity of Science and Technology [grant number 2020yjsjg03]; and the Key Laboratory of Storage of Agricultural Products, Ministry of Agricultural Products, Ministry of Agriculture and Rural Affairs [grant number Kf202208].

**Data Availability Statement:** The data presented in this study are available upon request from the corresponding author.

**Conflicts of Interest:** The authors declare no conflict of interest.

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
