# Peer review of "Coating of Layer-by-Layer Assembly Based on Chitosan and CMC: Emerging Alternative for Quality Maintenance of Citrus Fruit"

_horticulturae, doi:10.3390/horticulturae9060715_

Round 1
Reviewer 1 Report
The current study characterized the properties of CMC-chitosan layer-by-layer (LbL) films and the effects of chitosan and LbL coatings on the postharvest quality of a commercial mandarin variety in China.
The novelty of this study is not very high, since CMC-chitosan LbL coatings were already investigated in the past in various fruit types including citrus. Nonetheless, the current study adds a few new aspects and tested a new different citrus variety.
My comments are listed below:
1. The main problems in postharvest storage of mandarins are weight loss, decay and flavor deterioration. Therefore, the authors need to add missing data regarding the amounts of decay incidence and flavor evaluations.
2. M&M, line 149 – how were the coatings applied to the fruit? Was it by dipping, spraying, wiping, etc.?
3. Results, lines 243-244 – the following sentence is not clear: "the surface of the show rough surface …"
4. Fig. 1 – there is a problem with this figure, as the cut fruit after storage look to bright. It seems that the brightness was manipulated.
5. Results, line 307 – according to Fig. 5a the color values are not significant different.
6. Results, line 309 – the color angle data were not provided.
7. Fig. 5b – there seems to be a mistake with the firmness units. The firmness levels should be about 100 times lower between 0-50.
8. Fig. 5e – the TA data are very strange, as I never detected increase in TA values after storage.
9. Fig, 5f – I do not understand how ascorbic acid levels are significant different since the means are very similar and the standard errors are relatively large?
No comments
Author Response
Dear Reviewer,
Thank you for your letter dated June 1. We were pleased to know that our work was rated as potentially acceptable for publication in Journal, subject to adequate revision. We thank the reviewers for the time and effort that they have put into reviewing the previous version of the manuscript. Their suggestions have enabled us to improve our work. Based on the instructions provided in your letter. We uploaded the file of the revised manuscript. Accordingly, we have uploaded a copy of the original manuscript with all the changes highlighted by using the track changes mode in MS Word.
Appended to this letter is our point-by-point response to the comments raised by the reviewers. The comments are reproduced and our responses are given directly afterward.
We would like also to thank you for allowing us to resubmit a revised copy of the manuscript.
We hope that the revised manuscript is accepted for publication in Horticulturae.
Sincerely,
Chenyu Niu

Reviewer 2 Report
Dear authors, the submitted manuscript is interesting, however, I suggest the following aspects be considered:
Introduction Section
1.- Could you indicate the extension of shelf life by using multilayer protein/chitosan multilayer edible coating can extend the shelf life of fresh annona line 59-60
2.- Include the biographical citation of this description "Chitosan was found to provide mandarins with enhanced gloss and good firmness. However, the direct addition of chitosan-coated on the mandarin peel affects the normal gas exchange process" lines 61-62
3.- Include the bibliographic citation of this description "Therefore, the CMC was potentially used as the inner layer and the chitosan as the second layer of the outer layer with no direct contact with the peel. Electrostatic interactions between negatively charged carboxylic groups of CMC and positively charged ammonium groups of chitosan were utilized" lines 63-66.
4.- It is suggested to include examples of commercial waxes that are used to coat citrus fruits.
5.- It is suggested to revise the wording of lines 76-83, to adequately define the objective of the manuscript. Avoid including in this section the conclusion of the work
Methodology Section
1.-The following paragraph needs to be reviewed, line 104, the word "film" is repeated "The microstructure and film thickness of the film was determined by scanning electron microscopy".
Results Section
Properties characterization of the films
1.- In line 237-242 it is indicated that The LbL film has superior mechanical properties compared to the single chitosanone, relating the elasticity as a mechanical property (Table 1). However, the table does not show a statistically significant difference, you can review this?
Citrus Quality Attributes
Line 307 indicates that the L value of citrus in the CC group is higher than the L value of citrus in the CT group; however, Figure 5a does not show a significant difference between the groups or in the days.
3.- Why it was not considered adequate perform a microbiological assay to establish the effect of the films.
4.- It is necessary to include the conclusions of the manuscript.
Author Response

(The authors gave the same response as above.)

Round 2
Reviewer 1 Report
To my opinion this manuscript had to many flows and mistakes.
It could be accepted, but its definitely not the best manuscript I read.
Reviewer 2 Report
I thank you for your comments and recommendations. Success in future publications